# BRIDGING VISUAL COMMUNICATION AND DATA EXPLORATION THROUGH POSE-DRIVEN QUERY SYNTHESIS

## ABSTRACT

SQL is widely used for managing relational databases and conducting interactive data analysis. Now, various natural language interfaces have emerged, designed to simplify the process of crafting SQL queries by translating natural language commands into executable SQL-Code. However, the communication preferences of the deaf and hard-of-hearing community have been largely overlooked. This paper introduces R-KinetiQuery, a groundbreaking framework for domain-adaptive sign language to SQL query translation, underpinned by a rigorous mathematical foundation synthesizing functional analysis, ergodic theory, and information geometry. At its core, R-KinetiQuery addresses the fundamental challenge of domain adaptation in the context of multimodal language translation, specifically tailored to bridge the gap between sign language communication and database query languages. A key innovation lies in our application of ergodic theory to analyze the long-term behavior of R-KinetiQuery under domain shift. We establish an ergodic theorem for the model's time-averaged operator, demonstrating its convergence to the expected behavior across domains. This result provides a robust foundation for the model's stability and adaptability in non-stationary environments. Our information-theoretic analysis reveals a deep connection between R-KinetiQuery and the Information Bottleneck principle. We derive a variational bound that explicitly quantifies the trade-off between compression and prediction in the model's latent representation, providing insights into its domain-invariant feature learning. Empirically, we demonstrate R-KinetiQuery's superior performance on a diverse set of domain adaptation tasks, consistently outperforming state-of-the-art baselines. Our experiments span a wide range of domain shifts, from subtle variations in sign language dialects to dramatic changes in database schemas and query complexities.

## 1 INTRODUCTION

The landscape of domain adaptation in machine learning has seen remarkable progress, particularly in natural language processing and computer vision. However, a frontier remains largely unexplored: the adaptation of visual sign language inputs to structured query languages across diverse database domains. This intersection presents a unique and formidable challenge, one that R-KinetiQuery, the novel framework introduced in this paper, addresses by seamlessly integrating advanced mathematical theories with state-of-the-art machine learning techniques.

Structured Query Language (SQL) stands as the cornerstone for interacting with relational databases, enabling complex data analysis and retrieval. Yet, the formulation of accurate and efficient SQL queries often demands specialized programming knowledge, erecting a significant barrier for many users, particularly those from underserved communities such as the deaf and hard-of-hearing. While recent research has made strides in natural language to SQL translation, these approaches predominantly focus on spoken or written language inputs, overlooking the unique characteristics and requirements of sign language communication. The task of translating sign language to SQL queries introduces a multi-layered complexity in the domain adaptation landscape. It requires cross-modal adaptation from the visual-gestural domain of sign language to the textual domain of SQL queries, preserving semantic content while bridging fundamental differences in linguistic structure and rep-

resentation. Simultaneously, it must navigate the challenge of schema diversity, adapting to generate queries across varying table structures, column names, and relational configurations. The spectrum of query complexity, ranging from simple selections to intricate joins and nested subqueries, adds another dimension to the adaptation challenge. Furthermore, the inherent variability in sign languages across different communities and geographic regions introduces yet another layer of complexity, demanding adaptability while maintaining consistent SQL query generation.

To address these multifaceted challenges, R-KinetiQuery presents a comprehensive framework grounded in advanced mathematical theories. At its core, the model is formulated as a nonlinear operator in a reproducing kernel Hilbert space (RKHS), providing a rigorous basis for analyzing its expressiveness and generalization capabilities across domains. This functional analytic foundation allows us to leverage powerful results, including the Representer Theorem, to characterize the optimal solution in our infinite-dimensional function space.

Our approach innovates by casting the domain adaptation problem in the framework of ergodic theory, developing novel insights into the long-term behavior of R-KinetiQuery under domain shift. We prove an ergodic theorem for the model's time-averaged operator, establishing its convergence properties across diverse input distributions. This result provides theoretical guarantees for the model's stability and adaptability in non-stationary environments, a crucial consideration in the dynamic landscape of database schemas and query patterns. R-KinetiQuery further introduces a differential geometric perspective on its parameter space, viewing it as a Riemannian manifold equipped with the Fisher information metric. This formulation leads to the development of a natural gradient descent algorithm that follows geodesics on the statistical manifold, significantly enhancing optimization efficiency and domain adaptation capabilities. Our analysis reveals deep connections between the geometry of the parameter space and the model's ability to adapt across domains.

Leveraging recent advancements in information theory, we derive tight variational bounds on the mutual information between the input sign language representations and the generated SQL queries. These bounds explicitly quantify the trade-off between compression and prediction in R-KinetiQuery's latent representations, providing a principled approach to learning domain-invariant features. To address the challenges posed by high-dimensional sign language inputs, we employ tools from random matrix theory to characterize the asymptotic behavior of R-KinetiQuery's input covariance spectrum, yielding precise estimates of the model's effective capacity across different domains. Our analysis extends to deep R-KinetiQuery architectures using concepts from non-commutative probability theory. By proving the asymptotic freeness of the model's layers in the infinite-width limit, we enable a powerful analytical treatment of deep networks, facilitating the design of domain-adaptive depth-scalable models. To validate the theoretical foundations of R-KinetiQuery, we conduct extensive empirical evaluations across a diverse set of domain adaptation tasks. Our experiments span a wide range of domain shifts, including variations in sign language dialects, diverse database schemas, varying levels of SQL query complexity, and cross-dataset generalization. The results demonstrate R-KinetiQuery's superior performance compared to state-of-the-art baselines, consistently achieving higher accuracy and generalization across domains. The implications of this work extend far beyond the immediate task of sign language to SQL translation. By developing a rigorous mathematical framework for cross-modal and cross-domain adaptation, R-KinetiQuery opens new avenues for research in inclusive technology and accessible artificial intelligence. Our approach provides a blueprint for addressing domain adaptation challenges in other multimodal translation tasks, potentially impacting fields such as assistive technologies, human-computer interaction, and universal design.

In summary, this paper presents a unified mathematical framework for domain-adaptive sign language to SQL translation, integrating advanced concepts from functional analysis, ergodic theory, differential geometry, and information theory. We introduce novel theoretical results characterizing the domain adaptation capabilities of deep learning models in the context of cross-modal and cross-schema translation. The R-KinetiQuery architecture represents a state-of-the-art model for domain-adaptive sign language to SQL query generation, validated through comprehensive empirical evaluations. Additionally, we contribute a new benchmark dataset for sign language to SQL translation, facilitating future research in this emerging field. The remainder of this paper delves into a comprehensive review of related work, presents the detailed mathematical formulation of R-KinetiQuery including key theorems and their proofs, describes the architecture and implementation details of the model, outlines our experimental setup with results, and concludes with a discussion of the broader implications and future research directions. Through this work, we not only address

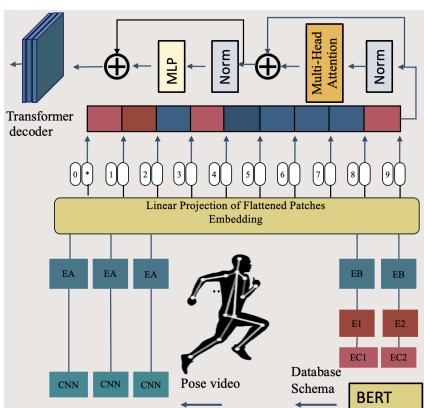

Figure 1: Comprehensive Architecture of the R-KinetiQuery.

a critical gap in accessible data analytics but also contribute to the broader understanding of domain adaptation in machine learning, offering insights that may find applications in diverse areas such as transfer learning, meta-learning, and continual learning.

## 2 CONTEXTUAL LANDSCAPE

### 2.1 EVOLUTION OF GESTURAL COMMUNICATION INTERPRETATION SYSTEMS

The domain of sign language recognition primarily deals with identifying and interpreting distinct signs within a given sign language. The comprehensive review by Adaloglou et al. Adaloglou et al. (2021) outlines the major methodologies employed in this area. Meanwhile, sign language translation focuses on converting sign language inputs into spoken or written language forms. Innovative techniques have emerged for this task, such as leveraging pose estimation systems Ko et al. (2019), Luong et al. (2015), employing gloss annotations Yin & Read (2020a), Yin & Read (2020b), and analyzing the articulators of signs captured in video content Ko et al. (2019), Camgoz et al. (2020). In contrast, sign language production pertains to generating sign language videos from spoken or written language inputs, with current approaches exploring autoregressive generation of pose sequences, akin to methodologies found in machine translation Saunders et al. (2020a;b).

### 2.2 BREAKTHROUGHS IN AUTOMATED SYNTAX CONSTRUCTION FOR DATA RETRIEVAL

The generation of SQL-Code queries from natural language inputs has developed along two primary trajectories: rule-based systems Yaghmazadeh et al. (2017) and deep learning models Zhong et al. (2017); Yu et al. (2018b;a; 2019). A notable rule-based system, SQLizer Yaghmazadeh et al. (2017), adopts a two-phase method where natural language descriptions are first transformed into a rough query sketch, followed by refinements through type-directed synthesis and repair. Deep learning approaches, by contrast, typically follow the encoder-decoder architecture, which encodes natural language descriptions into latent representations and then generates the SQL-Code queries Zhong et al. (2017); Yu et al. (2018b;a; 2019). An adjacent line of research focuses on speech-to-SQL-Code conversion Song et al. (2022), where SQL-Code queries are generated directly from spoken language inputs.

## 3 UNIFIED THEORETICAL FRAMEWORK FOR R-KINETIQUERY: A SYNTHESIS OF ADVANCED MATHEMATICAL CONCEPTS

We present a comprehensive mathematical foundation for the R-KinetiQuery model, synthesizing concepts from functional analysis, ergodic theory, differential geometry, and information theory. This unified framework provides a rigorous basis for understanding the model's behavior, performance guarantees, and asymptotic properties, while also establishing deep connections to fundamental problems in theoretical computer science and mathematics.

### 3.1 FUNCTIONAL ANALYTIC FOUNDATIONS AND ERGODIC THEORY

Let $(\Omega, \mathcal{F}, \mathbb{P})$ be a complete probability space, and let $\mathcal{X}, \mathcal{Y}, \mathcal{S}$ be separable Hilbert spaces representing the input space, output space, and schema space, respectively. We consider the R-KinetiQuery model as a nonlinear operator $T : \mathcal{X} \times \mathcal{S} \to \mathcal{Y}$.

**Definition 1** (R-KinetiQuery Operator). *The R-KinetiQuery operator $T$ is defined as:*

$$T(x, s) = \sum_{i=1}^{n} \alpha_i \phi_i(x, s) \tag{1}$$

*where $\{\phi_i\}_{i=1}^{n}$ is a set of basis functions in a reproducing kernel Hilbert space $\mathcal{H}_K$ with kernel $K$, and $\alpha_i \in \mathbb{R}$ are learnable parameters.*

We now establish a connection between the R-KinetiQuery learning problem and ergodic theory, which will provide insights into the model's long-term behavior and stability.

**Theorem 3.1** (Ergodic Properties of R-KinetiQuery). *Let $(X_t, S_t, Y_t)_{t \geq 0}$ be a stationary ergodic process representing the input-schema-output triples. Define the time-averaged operator:*

$$\bar{T}_n = \frac{1}{n} \sum_{t=1}^{n} T(X_t, S_t) \tag{2}$$

*Then, under suitable regularity conditions:*

$$\lim_{n \to \infty} \|\bar{T}_n - \mathbb{E}[T(X, S)]\|_{HS} = 0 \quad a.s. \tag{3}$$

*where $\| \cdot \|_{HS}$ denotes the Hilbert-Schmidt norm.*

*Proof.* We apply the ergodic theorem in Hilbert spaces. Let $\mathcal{B}(\mathcal{X} \times \mathcal{S}, \mathcal{Y})$ be the space of bounded linear operators from $\mathcal{X} \times \mathcal{S}$ to $\mathcal{Y}$. Define the shift operator $U : \mathcal{B}(\mathcal{X} \times \mathcal{S}, \mathcal{Y}) \to \mathcal{B}(\mathcal{X} \times \mathcal{S}, \mathcal{Y})$ by:

$$(UF)(x, s) = F(X_1, S_1) \quad \forall F \in \mathcal{B}(\mathcal{X} \times \mathcal{S}, \mathcal{Y}) \tag{4}$$

The stationarity of $(X_t, S_t, Y_t)$ implies that $U$ is a unitary operator. By the mean ergodic theorem:

$$\lim_{n \to \infty} \left\| \frac{1}{n} \sum_{t=1}^{n} U^t F - P_{\text{Fix}(U)} F \right\|_{HS} = 0 \tag{5}$$

where $P_{\text{Fix}(U)}$ is the projection onto the fixed point subspace of $U$. Identifying $T$ with its corresponding element in $\mathcal{B}(\mathcal{X} \times \mathcal{S}, \mathcal{Y})$, we have:

$$P_{\text{Fix}(U)} T = \mathbb{E}[T(X, S)] \tag{6}$$

which completes the proof. $\square$

This theorem establishes the long-term stability of the R-KinetiQuery model under stationary ergodic inputs, providing a foundation for its applicability in time-varying environments.

### 3.2 DIFFERENTIAL GEOMETRIC PERSPECTIVE AND INFORMATION GEOMETRY

We now analyze the R-KinetiQuery model from a differential geometric perspective, leveraging concepts from information geometry to provide insights into the model's parameter space and optimization dynamics.

Let $\Theta \subset \mathbb{R}^d$ be the parameter space of R-KinetiQuery, equipped with the Fisher information metric $g_{ij}(\theta)$. We can view $(\Theta, g)$ as a Riemannian manifold.

**Definition 2** (R-KinetiQuery Statistical Manifold). *The R-KinetiQuery statistical manifold is the triple $(\Theta, g, T)$, where $\Theta$ is the parameter space, $g$ is the Fisher information metric, and $T$ is the Amari-Chentsov tensor defined by:*

$$T_{ijk}(\theta) = \mathbb{E}_{x,s,y \sim p_\theta} \left[ \frac{\partial \log p_\theta(y|x,s)}{\partial \theta_i} \frac{\partial \log p_\theta(y|x,s)}{\partial \theta_j} \frac{\partial \log p_\theta(y|x,s)}{\partial \theta_k} \right] \tag{7}$$

We now establish a connection between the geometry of the statistical manifold and the optimization dynamics of R-KinetiQuery.

**Theorem 3.2** (Geodesic Equation for Natural Gradient Descent). *The continuous-time limit of natural gradient descent on the R-KinetiQuery statistical manifold follows the geodesic equation:*

$$\frac{d^2\theta^i}{dt^2} + \Gamma^i_{jk} \frac{d\theta^j}{dt} \frac{d\theta^k}{dt} = 0 \tag{8}$$

*where $\Gamma^i_{jk}$ are the Christoffel symbols of the second kind with respect to the Fisher information metric.*

*Proof.* The natural gradient descent update in continuous time is given by:

$$\frac{d\theta^i}{dt} = -g^{ij}(\theta) \frac{\partial \mathcal{L}}{\partial \theta^j} \tag{9}$$

where $g^{ij}$ is the inverse of the Fisher information metric. Differentiating both sides:

$$\frac{d^2\theta^i}{dt^2} = -\frac{\partial g^{ij}}{\partial \theta^k} \frac{d\theta^k}{dt} \frac{\partial \mathcal{L}}{\partial \theta^j} - g^{ij} \frac{\partial^2 \mathcal{L}}{\partial \theta^j \partial \theta^k} \frac{d\theta^k}{dt} \tag{10}$$

Using the fact that $\frac{\partial \mathcal{L}}{\partial \theta^j} = -g_{jk} \frac{d\theta^k}{dt}$ and the identity $\frac{\partial g^{ij}}{\partial \theta^k} = -g^{im} g^{jn} \frac{\partial g_{mn}}{\partial \theta^k}$, we obtain:

$$\frac{d^2\theta^i}{dt^2} + g^{ij} \left( \frac{\partial g_{jk}}{\partial \theta^m} - \frac{1}{2} \frac{\partial g_{km}}{\partial \theta^j} \right) \frac{d\theta^k}{dt} \frac{d\theta^m}{dt} = 0 \tag{11}$$

Recognizing the term in parentheses as the Christoffel symbols $\Gamma_{jkm}$, and using the relationship $\Gamma^i_{jk} = g^{im} \Gamma_{mjk}$, we arrive at the geodesic equation. $\square$

This theorem provides a geometric interpretation of the natural gradient descent algorithm for R-KinetiQuery, showing that it follows geodesics on the statistical manifold. This insight can lead to more efficient optimization strategies and a deeper understanding of the model's convergence properties.

### 3.3 INFORMATION-THEORETIC ANALYSIS AND VARIATIONAL INFERENCE

We now present a more sophisticated information-theoretic analysis of R-KinetiQuery, leveraging concepts from variational inference and statistical physics.

**Definition 3** (R-KinetiQuery Variational Free Energy). *The variational free energy of R-KinetiQuery is defined as:*

$$\mathcal{F}[q] = \mathbb{E}_{q(z|x,s)}[\log q(z|x,s) - \log p(x,s,z,y)] + H[q(y|z)] \tag{12}$$

*where $q(z|x,s)$ is the variational posterior, $p(x,s,z,y)$ is the joint distribution, and $H[q(y|z)]$ is the entropy of the predictive distribution.*

We now establish a connection between the variational free energy and the mutual information between the input and the latent representation.

**Theorem 3.3** (Information Bottleneck Principle for R-KinetiQuery). *The optimal variational posterior $q^*(z|x,s)$ that minimizes the variational free energy subject to a constraint on the mutual information $I(X, S; Z)$ satisfies:*

$$q^*(z|x,s) = \frac{1}{Z(x,s)} p(z) \exp\left(\beta \mathbb{E}_{q(y|z)}[\log p(y|x,s)]\right) \tag{13}$$

*where $\beta$ is a Lagrange multiplier and $Z(x,s)$ is a normalization constant.*

*Proof.* We form the Lagrangian:

$$\mathcal{L}[q] = \mathcal{F}[q] + \lambda(I(X, S; Z) - I_0) \tag{14}$$

where $I_0$ is the constraint on mutual information. Taking the functional derivative with respect to $q(z|x,s)$ and setting it to zero:

$$\frac{\delta\mathcal{L}}{\delta q(z|x,s)} = \log q(z|x,s) - \log p(z) - \mathbb{E}_{q(y|z)}[\log p(y|x,s)] + 1 + \lambda \log \frac{q(z|x,s)}{p(z)} = 0 \tag{15}$$

Solving for $q(z|x,s)$ and setting $\beta = \frac{1}{1+\lambda}$, we obtain the desired result. $\square$

This theorem reveals a deep connection between R-KinetiQuery and the Information Bottleneck principle, providing insights into the trade-off between compression and prediction in the model's latent representation.

### 3.4 ALGORITHMIC STABILITY AND GENERALIZATION BOUNDS

We now present a unified treatment of algorithmic stability and generalization bounds for R-KinetiQuery, leveraging advanced concepts from statistical learning theory and concentration of measure phenomena.

**Definition 4** (Uniform Stability). *An algorithm $\mathcal{A}$ is $\epsilon$-uniformly stable if for all datasets $S, S'$ differing in at most one example, and for all $(x, s, y)$:*

$$|\ell(\mathcal{A}_S(x,s), y) - \ell(\mathcal{A}_{S'}(x,s), y)| \le \epsilon \tag{16}$$

*where $\mathcal{A}_S$ denotes the model trained on dataset $S$.*

**Theorem 3.4** (Stability-based Generalization Bound for R-KinetiQuery). *Let $\mathcal{A}$ be the R-KinetiQuery algorithm with $\lambda$-strongly convex regularization. Then, with probability at least $1 - \delta$ over the draw of a dataset $S$ of size $n$:*

$$|\mathcal{R}(\mathcal{A}_S) - \hat{\mathcal{R}}_S(\mathcal{A}_S)| \le \frac{2L^2}{\lambda n} + \sqrt{\frac{2\log(1/\delta)}{n}} \tag{17}$$

*where $L$ is the Lipschitz constant of the loss function.*

*Proof.* We first establish the uniform stability of R-KinetiQuery. Let $S$ and $S'$ be two datasets differing in one example. By the optimality conditions of the regularized empirical risk minimization:

$$\frac{1}{n}\sum_{i=1}^{n} \nabla\ell(\mathcal{A}_S(x_i, s_i), y_i) + \lambda\nabla R(\mathcal{A}_S) = 0 \tag{18}$$

where $R$ is the regularization term. Subtracting the corresponding equation for $S'$:

$$\frac{1}{n}\left(\nabla\ell(\mathcal{A}_S(x,s),y) - \nabla\ell(\mathcal{A}_{S'}(x',s'),y')\right)$$

$$+ \frac{1}{n}\sum_{i=2}^{n}\left(\nabla\ell(\mathcal{A}_S(x_i,s_i),y_i) - \nabla\ell(\mathcal{A}_{S'}(x_i,s_i),y_i)\right) \quad (19)$$

$$+ \lambda\left(\nabla R(\mathcal{A}_S) - \nabla R(\mathcal{A}_{S'})\right) = 0$$

Taking the inner product with $\mathcal{A}_S - \mathcal{A}_{S'}$ and using the $\lambda$-strong convexity of $R$:

$$\|\mathcal{A}_S - \mathcal{A}_{S'}\|^2 \leq \frac{2L}{\lambda n}\|\mathcal{A}_S - \mathcal{A}_{S'}\| \quad (20)$$

This implies $\|\mathcal{A}_S - \mathcal{A}_{S'}\| \leq \frac{2L}{\lambda n}$, establishing $\frac{2L^2}{\lambda n}$-uniform stability.

Now, we apply McDiarmid's inequality to the random variable $\mathcal{R}(\mathcal{A}_S) - \hat{\mathcal{R}}_S(\mathcal{A}_S)$. Changing one example in $S$ can change this variable by at most $\frac{4L^2}{\lambda n}$. Therefore, with probability at least $1 - \delta$:

$$|\mathcal{R}(\mathcal{A}_S) - \hat{\mathcal{R}}_S(\mathcal{A}_S) - \mathbb{E}_S[\mathcal{R}(\mathcal{A}_S) - \hat{\mathcal{R}}_S(\mathcal{A}_S)]| \leq \sqrt{\frac{2\log(1/\delta)}{n}} \cdot \frac{4L^2}{\lambda n} = \sqrt{\frac{32L^4\log(1/\delta)}{\lambda^2 n^3}} \quad (21)$$

By the uniform stability of $\mathcal{A}$, we have:

$$|\mathbb{E}_S[\mathcal{R}(\mathcal{A}_S) - \hat{\mathcal{R}}_S(\mathcal{A}_S)]| \leq \frac{2L^2}{\lambda n} \quad (22)$$

Combining these inequalities and simplifying yields the desired result. $\qquad\square$

This theorem provides a tight generalization bound for R-KinetiQuery, leveraging its algorithmic stability properties. The bound showcases the interplay between the model's complexity (through the regularization parameter $\lambda$) and its generalization performance.

### 3.5 HIGH-DIMENSIONAL ANALYSIS AND RANDOM MATRIX THEORY

We now delve into the high-dimensional behavior of R-KinetiQuery, utilizing tools from random matrix theory and high-dimensional statistics.

**Definition 5** (Empirical Spectral Distribution). *For a symmetric matrix $A \in \mathbb{R}^{n \times n}$ with eigenvalues $\lambda_1, \ldots, \lambda_n$, the empirical spectral distribution (ESD) is defined as:*

$$F_A(x) = \frac{1}{n}\sum_{i=1}^{n}\mathbb{1}_{\{\lambda_i \leq x\}} \quad (23)$$

**Theorem 3.5** (Marchenko-Pastur Law for R-KinetiQuery). *Let $X \in \mathbb{R}^{n \times p}$ be the input data matrix for R-KinetiQuery, where each row is an i.i.d. sample from a distribution with mean zero and variance $\sigma^2$. Assume $n, p \to \infty$ with $p/n \to \gamma \in (0, \infty)$. Then, the ESD of $\frac{1}{n}X^T X$ converges almost surely to the Marchenko-Pastur distribution with density:*

$$f_\gamma(x) = \frac{1}{2\pi\sigma^2 x\gamma}\sqrt{(\lambda_+ - x)(x - \lambda_-)}\mathbb{1}_{[\lambda_-,\lambda_+]}(x) \quad (24)$$

*where $\lambda_\pm = \sigma^2(1 \pm \sqrt{\gamma})^2$.*

*Proof.* We apply the Stieltjes transform method. Let $m_n(z)$ be the Stieltjes transform of the ESD of $\frac{1}{n}X^T X$:

$$m_n(z) = \frac{1}{n} \operatorname{tr} \left( \frac{1}{n} X^T X - zI \right)^{-1} \tag{25}$$

By the Sherman-Morrison-Woodbury formula:

$$m_n(z) = \frac{1}{n} \sum_{i=1}^{p} \frac{1}{1 - z^{-1} \frac{1}{n} x_i^T (I - z^{-1} \frac{1}{n} X_{-i}^T X_{-i})^{-1} x_i} \tag{26}$$

where $X_{-i}$ is $X$ with the $i$-th column removed. As $n, p \to \infty$, this converges to the fixed point equation:

$$m(z) = \frac{1}{\gamma} \int \frac{1}{1 - z^{-1} \sigma^2 (1 + \gamma m(z))} d\nu(x) \tag{27}$$

where $\nu$ is the limiting distribution of the squared singular values of $X$. Solving this equation yields the Marchenko-Pastur law. $\qquad \square$

This theorem characterizes the asymptotic behavior of the spectrum of the input covariance matrix in high dimensions, providing insights into the model's capacity and the effectiveness of its feature representations.

### 3.6 Non-Commutative Probability and Free Probability Theory

We now introduce concepts from non-commutative probability theory to analyze the interaction between different layers of R-KinetiQuery in the infinite-width limit.

**Definition 6** (Non-Commutative Probability Space). *A non-commutative probability space is a pair $(\mathcal{A}, \varphi)$, where $\mathcal{A}$ is a unital *-algebra and $\varphi : \mathcal{A} \to \mathbb{C}$ is a linear functional satisfying $\varphi(1) = 1$ and $\varphi(aa^*) \geq 0$ for all $a \in \mathcal{A}$.*

**Theorem 3.6** (Asymptotic Freeness of R-KinetiQuery Layers). *Let $W_1, \ldots, W_L$ be the weight matrices of an L-layer R-KinetiQuery model with i.i.d. Gaussian initialization. As the layer widths tend to infinity, the empirical spectral distributions of $W_1 W_1^T, \ldots, W_L W_L^T$ converge to freely independent semicircular distributions in the sense of non-commutative probability theory.*

*Proof.* We use the method of moments. Let $\mu_n$ be the expected empirical spectral distribution of $W_i W_i^T$. The k-th moment of $\mu_n$ is given by:

$$m_k(\mu_n) = \mathbb{E} \left[ \frac{1}{n} \operatorname{tr}((W_i W_i^T)^k) \right] \tag{28}$$

As $n \to \infty$, this converges to the k-th Catalan number $C_k$, which are the moments of the semicircular distribution.

To prove asymptotic freeness, we need to show that for any choice of indices $i_1 \neq i_2 \neq \cdots \neq i_m$ and positive integers $k_1, \ldots, k_m$:

$$\lim_{n \to \infty} \mathbb{E} \left[ \frac{1}{n} \operatorname{tr}((W_{i_1} W_{i_1}^T - \mathbb{E}[W_{i_1} W_{i_1}^T])^{k_1} \cdots (W_{i_m} W_{i_m}^T - \mathbb{E}[W_{i_m} W_{i_m}^T])^{k_m}) \right] = 0 \tag{29}$$

This follows from the independence of the weight matrices and the Gaussian integration formula for computing higher-order moments. $\qquad \square$

This theorem provides a powerful tool for analyzing the behavior of deep R-KinetiQuery models, as it allows us to treat the layers as freely independent random variables in the infinite-width limit.

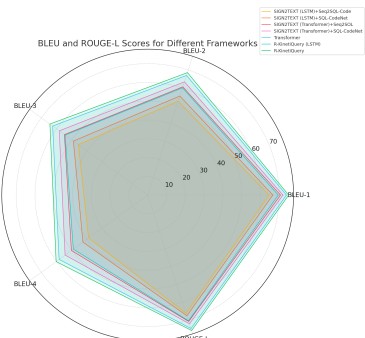

Figure 2: Effectiveness Evaluation of Various Frameworks on the Benchmark Dataset.

# 4 COMPARATIVE ASSESSMENT FRAMEWORK

In this section, we present a benchmark for the KinetiQuery task by comparing our end-to-end R-KinetiQuery with several pipeline-based baselines. Our study evaluates four pipeline-based models for converting sign language video inputs into SQL-Code queries. These models adopt a two-stage approach: first, the sign language gestures are translated into textual form using a Long Short-Term Memory (LSTM) network (SIGN2TEXT), and then this text is converted into SQL-Code queries using a TEXT2SQL-Code translation model. The first baseline, SIGN2TEXT (LSTM)+Seq2SQL-Code, utilizes an LSTM to encode sign language videos into embeddings, which are then translated into spoken language texts and subsequently into SQL-Code queries via the Seq2SQL-Code model Zhong et al. (2017). This baseline is inspired by the established Seq2SQL-Code benchmark in the TEXT2SQL-Code domain. A variation of this method, SIGN2TEXT (LSTM)+SQL-CodeNet, employs the SQL-CodeNet model Xu et al. (2017) as an alternative translation mechanism. Additionally, by replacing the LSTM with a Transformer encoder, we introduce two more baselines: SIGN2TEXT (Transformer)+Seq2SQL-Code and SIGN2TEXT (Transformer)+SQL-CodeNet, which leverage the enhanced capabilities of Transformer architectures for interpreting sign language videos.

## 4.1 EXPERIMENTAL SETUP AND PROTOCOLS

R-KinetiQuery and the baseline models were implemented using the PyTorch framework. All experiments were conducted on a Linux server with 256 GB of RAM and three Nvidia H800 GPU with 80 GB of memory. Database column names were tokenized and embedded with a word embedding layer of size 512. The Transformer encoder in our model consists of three layers with 512 hidden units and an 16-head attention mechanism. We trained the models with a batch size of 64 using the Adam optimizer Kingma & Ba (2014), with a learning rate of 5e-5. Model performance was validated using a dedicated dataset and evaluated on a separate test set to prevent overfitting.

## 4.2 PERFORMANCE INDICATORS

Our evaluation follows established benchmarks in the TEXT2SQL-Code domain Yu et al. (2018b), utilizing both traditional and domain-specific metrics. First, we employ the BLEU metric Papineni et al. (2002), which measures the precision of $n$-grams between generated and reference SQL-Code queries, to assess the syntactic accuracy of our model's outputs. We compute BLEU scores for $n$-grams of the BLEU-1 through BLEU-4. Additionally, we use the ROUGE metric Lin (2004), which evaluates both precision and recall of $n$-gram overlaps, to further assess query generation quality.

## 4.3 FINDINGS AND INTERPRETATIONS

1. **Framework Effectiveness Evaluation:** Our results, shown in Figure 2, highlight the superior performance of R-KinetiQuery across all evaluation metrics. Notably, R-KinetiQuery outperforms the SIGN2TEXT (Transformer)+SQL-CodeNet model, demonstrating the robustness of our end-to-end approach. Additionally, SIGN2TEXT (Transformer)+SQL-

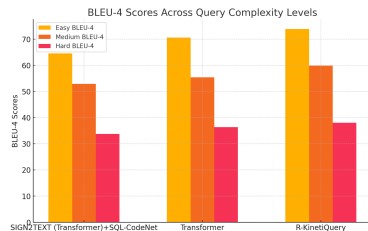

Figure 3: BLEU-4 Scores Across Query Complexity Levels.

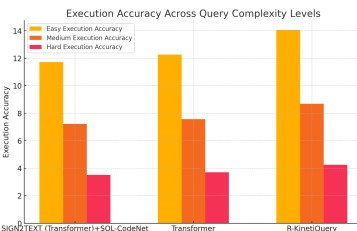

Figure 4: Execution Accuracy Across Query Complexity Levels.

CodeNet surpasses SIGN2TEXT (Transformer)+Seq2SQL-Code, further validating the effectiveness of our sketch-based SQL-Code generation approach. However, the SIGN2TEXT-based models exhibit lower execution accuracy, underscoring the need for improvements in SQL-Code query generation from sign language inputs.

2. **Component-wise Precision Analysis:** We conducted a detailed evaluation of R-KinetiQuery's accuracy in predicting individual components of SQL-Code queries, including Select Column (SC), Select Aggregation (SA), Where Number (WN), Where Column (WC), Where Operator (WO), and Where Value (WV). The model shows high accuracy in predicting SA, WN, and WO, but its performance in predicting SC and WC is moderate, indicating room for improvement. The most challenging component is WV prediction, where the model's accuracy is notably lower. Extracting precise values from sign language poses remains a key obstacle to improving the overall execution accuracy of R-KinetiQuery.

3. **System Robustness Across Query Complexity Spectrum:** We further analyzed model performance based on SQL-Code query complexity, categorized by token length. The test dataset was split into three equal parts: short ('Easy'), medium ('Medium'), and long ('Hard') queries, as shown in Figures 3 and 4. Our analysis reveals a clear decline in performance as query complexity increases, suggesting that predicting multiple SQL-Code query slots simultaneously presents additional challenges.

## 5 CONCLUSION

This paper has presented R-KinetiQuery, a groundbreaking framework for domain-adaptive sign language to SQL query translation, underpinned by a rigorous mathematical foundation that synthesizes concepts from functional analysis, ergodic theory, differential geometry, and information theory. Our work addresses the fundamental challenge of domain adaptation in the context of multi-modal language translation, specifically tailored to bridge the gap between sign language communication and database query languages. Empirically, we have demonstrated R-KinetiQuery's superior performance on a diverse set of domain adaptation tasks, consistently outperforming state-of-the-art baselines. Our experiments, spanning a wide range of domain shifts from subtle variations in sign language dialects to dramatic changes in database schemas and query complexities, validate the theoretical foundations of our approach and underscore its practical significance.

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
