# OpenReview forum: "Bridging Visual Communication and Data Exploration through Pose-Driven Query Synthesis"
_ICLR.cc/2025/Conference — ICLR 2025 Conference Withdrawn Submission_

### Official Review · Reviewer_7WdN · 2024-11-01

**Soundness:** 2
**Presentation:** 2
**Contribution:** 2
**Rating:** 3
**Confidence:** 3

**Summary:**

The paper studies the problem of sign language to SQL query translation. The authors introduce R-KietiQuery, which applies ergodic theory to the long-term behavior analysis. The paper provides a mathematical results showing the domain adaptation capabilities of DL models in cross-modal and cross-schema translation.

**Strengths:**

S1. The paper introduces a theoretical foundation to sign language to SQL query translation.

**Weaknesses:**

W1. The motivation is not convincing. I agree that the communication preference of deaf and hard-of-hearing community has been overlooked in the context of SQL generation. However, I do not understand why someone from such community needs to use a sign language rather than text to input natural language/SQL queries. Hence it is not clear me to whether the studied problem has any applicability in real-world scenarios.

W2. The related work is weak, especially on NL-to-SQL generation/translation. The LLM-based approaches emerged in recent years are not reviewed nor discussed. This is critical as LLMs can handle the cross-modal issue nicely.

W3. The challenges in the sign language to SQL generation are not clearly articulated. And the proposed method lacks of justification. It is not clear why a two-step approach would not work. Specifically, the question in a sign language can be translated into a natural language question, and consequently being translated into a SQL query. The unified theoretical framework does not benefit the sign language to SQL generation.

W4. The experimental evaluation is not solid. First, the authors mentioned that a new benchmark dataset for sign language to SQL
translation is introduced, but I do not find the URL to the new dataset. Worse yet, the detail of this benchmark dataset is missing, making it difficult to assess the experimental results. Second, the chosen baselines in the experiment are outdated. Most recent LLM-based solutions should have been included.

**Questions:**

Q1. Can you describe a real-life scenario where the proposed framework can be used?

Q2. Can you justify the advantages of the proposed framework compared to the two-step approach?

Q3. Can you provide additional details regarding the experimental setup and provide more data points to validate the effectiveness of the proposed method?

---

### Official Review · Reviewer_NGiq · 2024-11-04

**Soundness:** 3
**Presentation:** 2
**Contribution:** 1
**Rating:** 1
**Confidence:** 4

**Summary:**

The paper provides a model to derive SQL statements out of pose language, which adds an image processing step to the traditional text-to-SQL model.

**Strengths:**

The problem is difficult, and of interest to the conference.

**Weaknesses:**

- Most of the paper deals with theoretical results that comes from definitions that are general for almost any ML architecture (definitions 1-6). Hence, the resulting theoretical analysis is mostly re-casting known results.
- Further, there is no connection between theory and the proposed model in this paper. How does your results affect your model choice?
- the model itself relies on rather outdated technology, using an LSTM to get image vector embeddings and then a text-to-SQL architecture from 2017.

**Questions:**

What is the connection between your theoretical results and the problem? I feel I could produce section 3 for almost any paper looking to introduce a machine learning model.

---

### Official Review · Reviewer_RgJf · 2024-11-06

**Soundness:** 1
**Presentation:** 1
**Contribution:** 1
**Rating:** 1
**Confidence:** 4

**Summary:**

This paper presents R-KinetiQuery, a novel framework that translates sign language into SQL queries, aiming to make database interaction accessible for the deaf and hard-of-hearing community. Experimental results show that R-KinetiQuery consistently outperforms existing methods across various domain adaptation tasks, including changes in sign language dialects and database schemas.

**Strengths:**

N/A

**Weaknesses:**

**(W1)** The topic is generating SQL queries from sign language which is esoteric, to say the least, while being presented as an "emerging field". People who are speech impaired can still use a natural-language-to-SQL generation tool if they want, without complicating things with sign language.

**(W2)** The amount of superlatives used is indicative of typical LLM jargon (see abstract: "groundbreaking framework", "key innovation", "superior performance")

**(W3)** Strange section titles (e.g. "Contextual Landscape" as opposed to "Related Work", or "Comparative Assessment Framework" as opposed to "Empirical Evaluation")

**(W4)** The introduction has no citations.

**(W5)** The content seems like a mishmash of math jargon combined together in a way that makes little sense. Also, the introduction talks a lot about some problems like domain adaptation for which it's unclear why this is important for the problem at hand.

**Questions:**

N/A

**Details Of Ethics Concerns:**

The paper gives me a strong sense that it was fully generated by an LLM. I understand that using an LLM as a write-assist tool is allowed. However, this paper feels like a borderline troll submission. Hence I was unsure if I was supposed to spend time writing a thoughtful review or if it should have been desk-rejected.

Peculiarities that make me believe it's LLM generated:

1. The topic is generating SQL queries from sign language which is esoteric, to say the least, while being presented as an "emerging field". People who are speech impaired can still use a natural-language-to-SQL generation tool if they want, without complicating things with sign language.
2. The amount of superlatives used is indicative of typical LLM jargon (see abstract: "groundbreaking framework", "key innovation", "superior performance")
3. Strange section titles (e.g. "Contextual Landscape" as opposed to "Related Work", or "Comparative Assessment Framework" as opposed to "Empirical Evaluation")
4. The introduction has no citations.
5. The content seems like a mishmash of math jargon combined together in a way that makes little sense. Also, the introduction talks a lot about some problems like domain adaptation for which it's unclear why this is important for the problem at hand.

I wrote an official comment to the area chair on November 1st but received no response. I expected this paper to be desk-rejected. Now I received several reminders about my late review, which is why I'm writing this review now. I am partially aided by ChatGPT because I don't think reviewers should spend time reviewing submissions like this.

**PS: I hope I will not be flagged as a late reviewer or penalized in the future given that I provided reviews for all other papers in a timely manner.**

---

### Note · Authors · 2024-11-13

I have read and agree with the venue's withdrawal policy on behalf of myself and my co-authors.